# The Role of Type VI Collagen in Alveolar Bone

**DOI:** 10.3390/ijms232214347

**Published:** 2022-11-18

**Authors:** Taishi Komori, Hai Pham, Priyam Jani, Sienna Perry, Yan Wang, Tina M. Kilts, Li Li, Marian F. Young

**Affiliations:** 1Molecular Biology of Bones and Teeth Section, National Institute of Dental and Craniofacial Research, Department of Health and Human Services, National Institutes of Health, Bethesda, MD 20892, USA; 2Faculty of Dentistry, Hai Phong University of Medicine and Pharmacy, Haiphong 180000, Vietnam; 3Mass Spectrometry Facility, National Institute of Dental and Craniofacial Research, Department of Health and Human Services, National Institutes of Health, Bethesda, MD 20892, USA

**Keywords:** type VI collagen, µCT analysis, proteome analysis, alveolar bone

## Abstract

Many studies have been conducted to elucidate the role of Type VI collagen in muscle and tendon, however, its role in oral tissues remains unclear. In this study, an α2(VI) deficient mouse (*Col6α2*-KO) model was used to examine the role of Type VI collagen in oral tissues. Tissue volume and mineral density were measured in oral tissues by µCT. Proteome analysis was performed using protein extracted from alveolar bone. In addition, alveolar bone was evaluated with a periodontitis induced model. µCT analysis showed the *Col6α2*-KO mice had less volume of alveolar bone, dentin and dental pulp, while the width of periodontal ligament (PDL) was greater than WT. The mineral density in alveolar bone and dentin were elevated in *Col6α2*-KO mice compared with WT. Our proteome analysis showed significant changes in proteins related to ECM organization and elevation of proteins associated with biomineralization in the *Col6α2*-KO mice. In induced periodontitis, *Col6α2*-KO mice had greater alveolar bone loss compared with WT. In conclusion, Type VI collagen has a role in controlling biomineralization in alveolar bone and that changes in the ECM of alveolar bone could be associated with greater bone loss due to periodontitis.

## 1. Introduction

Type VI collagen is a microfibrillar collagen found in many connective tissues including muscle, tendon, skin, bone and cartilage [1,2,3,4]. It is composed of three alpha chains, and made mainly of α1(VI), α2(VI) and α3(VI) chains [5]. Three other alpha chains have also been identified that are similar to α3 (α4(VI), α5(VI), α6(VI)) [6]. Human mutations in the Type VI collagen alpha chains have been found in patients with muscle disorders known as Type VI collagen-related myopathies that include Ullrich congenital muscular dystrophy (UCMD) and Bethlem myopathy (BM) [2,7,8]. To understand the mechanistic basis for these diseases and to develop new treatments, many studies have focused on the roles of type VI collagen in muscle and tendon, however, less is known about its role in oral tissues.

The extracellular matrix (ECM) has an important role in providing a scaffold, facilitates interaction between cell–cell or cell-signaling molecules and regulates growth and homeostasis in tissues. Type VI collagen is known to be an important factor in organizing ECM structure by interacting with other ECM molecules. These molecules include Type I [9], Type II [10], Type IV [11] collagens, perlecan [12], decorin [13], biglycan [14] and fibronectin [15], as well as integrins [16] and the cell-surface proteoglycan NG2 [17]. Given that Type VI collagen has bridging functions with numerous ECM molecules, it is proposed that it integrates the ECM structure and regulates homeostasis in tissues. Previous reports using a Type VI collagen-deficient mouse model showed that they have defective tendons with abnormal fibroblast shape and greater levels of thin fibers resulting in weak mechanical properties [18,19]. Other studies show Type VI collagen is an important component of the stem cell niche in muscle and can regulate self-renewal of muscle stem cells and, subsequently, muscle regeneration [20,21]. Type VI collagen also appears to contribute to inflammation in adipose and nervous tissue [22,23]. Taken together, it is concluded that Type VI collagen has numerous roles in many tissues by controlling its ECM components.

Oral tissues are complex consisting of mineralized tissues including teeth and alveolar bone and non-mineralized tissues including gingiva and the periodontal ligament (PDL), all of which support the structure and function of the mandible and maxilla. Tooth tissues consist of enamel at the surface covering the dentin and dental pulp underneath. Healthy alveolar bone supports the teeth and enables them to be functional in chewing and for stabilizing the jaw. Although alveolar bone is composed primarily of hydroxyapatite, its ECM scaffold is an important factor in controlling hydroxyapatite deposition. Collagen is the main organic component of bone, and its distribution and cross-linking make important contributions to bone quality.

Periodontitis is the most common disease in oral tissues and presents with chronic inflammation leading to alveolar bone loss. Past studies show many factors are involved in the regulation of periodontitis including bone quality, chewing forces, immune responses and bacterial activity, however, little is known for the role of ECM in periodontitis.

The current study was aimed to examine changes in oral tissues using α2(VI) deficient mouse (*Col6α2*-KO). To examine the potential networking function of Type VI collagen in alveolar bone, a proteome analysis was performed. In addition, bone loss with induced periodontitis was used to identify whether Type VI collagen deficiency affects bone loss progression due to induced periodontitis.

## 2. Results

### 2.1. Type VI Collagen Is Expressed in Oral Tissues

Oral tissues consist of mineralized alveolar bone (AB), teeth that contain dental pulp (DP), as well as non-mineralized tissues including the gingiva (Gn) and the periodontal ligament (PDL) (Figure 1A,B). To examine the expression of Type VI collagen in oral tissues, we first performed immunohistochemistry for Type VI collagen using 3-month-old WT mice. The immunostaining showed the unique expression pattern of Type VI collagen in non-mineralized tissues including the PDL (Figure 1a’,c’, yellow arrowheads), the bone marrow in alveolar bone (Figure 1a’, green arrowheads), the dental pulp (Figure 1b’) and in the connective tissue (CT) of the gingiva (Figure 1d’, blue arrowheads). To further investigate the protein composition of the mineralized tissue, alveolar bone was dissected, protein was extracted from 6-week-old mice, and then subjected to proteome analysis. In mice teeth, cementum tissue covers the root deposits with aging, making it difficult to pull out the tooth intact. To avoid contamination from other tooth tissues, we used the 6-week old mice. Our proteome analysis identified α1(VI), α2(VI), α3(VI) and α5(VI) were abundantly expressed (Figure 1C). The small size and intricate shape of the enamel and dentin precluded a similar proteomic analysis in these latter tissues.

### 2.2. Col6α2-KO Mice Have Compromised Oral Tissues

Since Type VI collagen is expressed in numerous oral tissues, we used an α2(VI) chain deficient (*Col6α2*-KO) mouse model that is globally unable to secrete any Type VI collagen into its ECM [24]. In this study, µCT was used to measure the volume of the alveolar bone and the nature of the tooth tissue using software that allowed the separation of enamel, dentin and dental pulp. In our analysis, we also measured the width of the PDL. An illustration showing tissue measurements of the µCT analysis is shown in Figure 2A. Quantification of the µCT data showed that compared with WT, the *Col6α2*-KO mice had significantly less alveolar bone volume (*p* < 0.005), less dentin volume (*p* < 0.005) and less dental pulp volume (*p* < 0.05), however, there was no significant differences in the enamel (Figure 2C–F). When the width of PDL was measured in a ROI using the mesial root of 1st molar (Figure 2B), we found the width of the PDL was greater in the *Col6a2*-KO mice (*p* < 0.05) (Figure 2G). There were no significant differences in the ratio in the volume of the overall tooth to alveolar bone in *Col6a2*-KO compared to WT mice. To further analyze the nature of the mineralization status of the alveolar bone, enamel and dentin, the mineral density was measured. A heatmap showing the mineral density of oral tissues is shown in Figure 3A. Quantification of the mineral density showed *Col6α2*-KO mice had greater mineral density in alveolar bone (*p* < 0.01) and dentin (*p* < 0.05) but no significant changes in enamel (Figure 3B–D).

### 2.3. Type VI Collagen Affects ECM Molecules in the Alveolar Bone

Since Type VI collagen protein expression was found in alveolar bone, we next performed proteome analysis to try to further decipher its molecular functions at a developmental stage active in mineral apposition (6-weeks). From our proteome analysis using 6-week-old mice, we found significantly less expression of α1(VI), α2(VI), α3(VI) and α5(VI) in the alveolar bone of the *Col6α2*-KO mice compared with WT mice (Figure 4A, red box). Other collagens affected by α2(VI) deficiency were α2(IV), α1(XVI) and α1(XXII), that were significantly less in α2(VI) deficient alveolar bone. Our proteome data also identified 287 differentially expressed proteins (DEPs) (*p*-value < 0.05). Gene Ontology enrichment analysis was carried out to understand the potential role of Type VI collagen on biological functions. Type VI collagen has suggested functions in organizing the ECM structure, and as expected, proteins related to Extracellular Matrix Organization (GO: 0030198) were enriched. These included 19 proteins shown in yellow in Figure 4B. Interestingly, there were several important molecules previously implicated in Biomineralization (GO: 0110148) including enamelin (Enam), Ameloblastin (Ambn), Amelogenin (Amelx), Dentin sialophosphoprotein (Dspp), alkaline phosphatase (Alpl) and Fibroblast growth factor receptor 1 (Fgfr1), all of which were upregulated in α2(VI) deficient alveolar bone (Figure 4B, green). The proteins marked half color of yellow and green belong to both GO terms.

### 2.4. Col6α2-KO Mice Have More Bone Resorption in an Induced Model of Periodontitis Compared with WT

To examine whether Type VI collagen deficiency affects induced bone loss, we used a ligature procedure which is a well-accepted model to induce periodontitis. In this experiment, a thin piece of string was tied around the right maxillary 2nd molar (ligature side) in WT and *Col6α2*-KO mice, and then bone parameters were measured after 3 days. This procedure induced the accumulation of bacteria which then caused bone resorption. The left maxillary side without a ligature served as a control. Bone loss was measured by µCT. To evaluate vertical bone loss, the distance between the cemento-enamel junction (CEJ) to the alveolar bone crest (AB) was measured as shown in Figure 5A and the difference in CEJ-AB distance in the control side and the ligature side were compared. In addition, the bone volume was measured to determine the amount of alveolar bone loss in 3D. Our results showed that with periodontitis both the CEJ-AB distance (Figure 5B) and % volume of bone loss was significantly greater in *Col6α2*-KO compared with WT mice (Figure 5C).

## 3. Discussion

Previous studies show that Type VI collagen regulates collagen fibril assembly, ultimately affecting fiber thickness [18,19], as well as the function of stem cells for regeneration of muscle and tendon [20,21]. Although Type VI collagen is found in many tissues, little is known about its role in oral tissues. This current study showed for the first time that the oral tissues in *Col6α2*-KO mice are abnormal and have greater mineral density in alveolar bone and dentin compared with WT mice. Proteome analysis of the alveolar bone showed that deficiency in Type VI collagen resulted in changes in proteins related to ECM organization and biomineralization and that this could be the basis for the abnormal volume and mineral density in alveolar bone found in the *Col6α2*-KO mice. We show the alveolar bone affected by the loss of Type VI collagen has increased bone loss with induced periodontitis.

There are few reports that examined the volume of oral tissues using µCT. Mice deficient in *osteopontin*, an ECM molecule that regulates bone development, have greater alveolar bone and dentin volume and less dental pulp and PDL volume compared with WT mice [25]. Other work using a mutant mouse model of X-linked Hypophosphatemia (XLH) with a deficiency in *phex,* the gene responsible for phosphate regulation, showed that they have less enamel and dentin with no significant change in alveolar bone volume [26]. He Xu et al., showed mice deficient in *crtap, a* cartilage-associated protein that is associated with post-translational modification of Type I collagen, as well as mice with osteogenesis imperfecta Type VII due to loss of *Col1a2* (oim) have less alveolar bone and dentin volume and greater PDL volume [27]. Here, we showed that mice deficient in *Col6α2* have defects in the volume of oral tissues suggesting a regulatory role of Type VI collagen in the development or homeostasis of oral tissues. In our work, we used 3-month-old mice at which time the growth of oral tissues is completed. Further studies are needed using tissues from earlier stages to fully understand if and how Type VI collagen could regulate the development and final formation of oral tissues. The mineral density of alveolar bone was also evaluated in these reports and showed greater mineral density in the alveolar bone of *osteopontin* knock-out mice and less mineral density in the alveolar bone in XLH deficient and oim mice, the latter of which is a model of OI type VII described earlier. Previous work from our lab showed that *fibromodulin* knock-out mice have greater mineral density in their alveolar bone compared with WT mice [28]. In this study, we showed *Col6α2*-KO mice also had greater mineral density in alveolar bone and dentin compared with WT mice. From these previous reports and the current study, we propose that ECM molecules including osteopontin, fibromodulin, Type I and Type VI collagen, as well as proteins implicated in phosphate regulation, could control mineral apposition in alveolar bone. However, more studies are necessary to understand their mechanism of action.

The orchestration of bone biomineralization is complex and appears to be regulated by non-collagenous proteins (NCPs), including glycoproteins and proteoglycans that interact closely with inorganic calcium and phosphate ions to control the deposition of hydroxyapatite (HA) on extracellularly secreted collagen fibers. Small integrin-binding ligand N-linked glycoprotein (SIBLINGs) are some of the NCPs which are important for mineralization in bone and dentin and include Dmp1, Mepe, Opn which inhibits calcification and Dspp1 which enhances calcification. Our proteome analysis showed the levels of Dmp1, Mepe and Opn were not changed in the absence of Type VI collagen, however, the expression of Dspp1 was upregulated in the *Col6α2*-KO mice. Interestingly, the levels of the enamel matrix proteins (EMPs) including Amelx, Ambn and Enam were upregulated in the alveolar bone of the *Col6α2*-KO mice. The EMPs have been identified in enamel matrix and considered important factors for the regulation of enamel mineralization. Recent studies showed EMPs are also expressed in non-enamel mineralized tissues and can increase osteogenesis in bone marrow stem cells (BMSCs) [29,30,31,32]. Amelx and Ambn were also previously found in alveolar bone [33]. Considering the role of EMPs in biomineralization, it is possible their upregulation could be the mechanistic basis of the increased mineral density we found in the *Col6α2*-KO mice, however, additional studies need to be performed to understand their exact function in regulating mineral density in alveolar bone. An important next step will be to carry out Western blotting to confirm the upregulation of proteins related to biomineralization including the SIBLINGs and the EMPs.

Our observation that there is greater mineral density in alveolar bone in *Col6α2*-KO compared with WT mice is different from our previous findings in trabecular bone in long bone where we found less mineral density in *Col6α2*-KO compared with WT mice. While the reason for this difference is not known, it is widely accepted that alveolar bone and trabecular bone have different growth mechanisms. Specifically, orofacial bone is neural crest derived and long bones are from mesoderm. It will be interesting in the future to determine how and why Type VI collagen has unique tissue-specific effects depending on tissue location and developmental origin. The expression of the EMPs in alveolar bone could be involved the differential effect of Type VI collagen on long bone compared to alveolar bone.

Three genetically distinct Type VI collagen alpha chains, α1(VI), α2(VI) and α3(VI) were originally identified. However, later studies uncovered that there are three additional alpha chains, α4(VI), α5(VI) and α6(VI) which have homology with α3(VI). In our proteome analysis using protein extracted from alveolar bone α5(VI) was identified. Although it is known that the α5(VI) protein is expressed more widely in mouse compared with human tissues, the reason for the restricted expression in humans is not clear. In the assembly of Type VI collagen, it is known that α1 (VI), α2 (VI) and α3(VI) chains are the basic units that make up the [α1, α2, α3] heterotrimers with a 1:1:1 ratio. As for α5 (VI), it is speculated that because of its homology it can substitute for the α3(VI) chain and generate [α1, α2, α5] heterotrimers [34]. Although we do not know the exact function of α5(VI) in alveolar bone, its presence provides important insights about possible novel roles for these newly identified alpha chains of Type VI collagen.

Periodontitis causes alveolar bone loss due to chronic inflammation. The exact mechanisms regulating bone loss during periodontitis are complex because many factors such as bone quality, chewing force, immune response are involved. In this study, we showed greater mineral density in alveolar bone in *Col6α2*-KO mice accompanied by a significant change in proteins related to ECM organization. The structure of collagen in the bone has been recognized as an important factor in bone quality. Although Type I collagen detected in the proteome analysis did not have altered expression in *Col6α2*-KO and we did not perform collagen fibers analysis in the alveolar bone, we speculate that the altered properties we found in alveolar bone in the α2(VI) deficient mice, such as greater mineral density and disorganized ECM organization could be a factor for increased bone loss from periodontitis, however, other aspects of the function in Type VI collagen need to be studied in the future. As shown in Figure 1d’, Type VI collagen is expressed in the gingiva. Previous reports using single cell RNA analysis show there is a change in the cell populations in the gingiva from healthy controls and periodontitis patients [35]. When we analyzed the publicly available data bases from this study, we found that Type VI collagen expression was upregulated in the gingiva from periodontitis patients, suggesting Type VI collagen is secreted to protect tissues from inflammation or to induce regeneration of tissues subject to destruction. The PDL is another oral tissue expressing Type VI collagen, and in this context, it is interesting to note that the PDL of *Col6α2*-KO mice has weak mechanical properties. This could explain why the width of the PDL in the *Col6α2*-KO mice is wider and could possibly contribute to the bone loss that occurs during periodontitis.

The limitation of this study is that an analysis is needed to understand the role of Type VI collagen in bone loss from periodontitis. Given that Type VI collagen is expressed in the gingiva and PDL, it might play protective roles there in ameliorating periodontitis. Further, previous studies showing its role in modulating macrophage support this possibility [22,23]. To address this point, specific studies on gingiva and the PDL are needed.

The present study was intended to show the expression of Type VI collagen in oral tissues and determine possible functions in alveolar bone mineralization. Our results suggest that Type VI collagen regulates biomineralization in alveolar bone and altered alveolar bone properties could lead to increased bone loss during induced periodontitis.

## 4. Materials and Methods

### 4.1. Animal Experiments

The *Col6a2*-KO mouse strain used for this research project was a kind gift from Carsten Bonnemann (National Institute of Neurological Disorders and Stroke, NIH). Animals were housed under standard conditions (55% humidity, 12 h day night cycle, standard chow and free access to water) following the guidelines and approval of The National Institutes of Dental and Craniofacial Research Animal Care and Use Committee (protocol #21-1067). A total of 40 WT and *Col6a2*-KO mice were used in this study.

### 4.2. Micro-Computed Tomography (µCT)

For µCT analysis, mice were euthanized, and mandibles were dissected from 3-month-old WT and *Col6α2*-KO and fixed for 24 h at room temperature in Z-fix (Anatech, LTD, Battle Creek, MI, USA) and then stored in 70% ethanol at 4 °C. The 3-D reconstruction image of mandibles were acquired using micro-CT system (µCT 50, Scanco Medical AG, Bassersdorf, Switzerland) with the following parameters: 70 kV X-ray source voltage, 85 µA of intensity/beam current, power at 6 W and integration time at 300 ms. The image resolution was 6 µm. The 3D mandible images were rendered, and Bone volume (BV) and mineral density were measured using AnalyzePro software (AnalyzeDirect, Overland Park, KS, USA). To measure the PDL width, a region of interest (ROI) was determined using the cemento-enamel junction (CEJ) and the apex of the root. The middle position of root was calculated from the CEJ and the apex of the root, and 25 slices from the middle position to the CEJ side and 25 slices from the middle position to the apex side was used for the ROI (1 slice = 6 µm).

### 4.3. Immunohistochemistry

The dissected mandibles from 3-month-old mice were fixed in Z-fix for 24 h, rinsed with PBS overnight and decalcified with 10% EDTA for 14 days. Samples were then washed and dehydrated through a graded ethanol series and xylene before paraffin embedding. The sections were cut at 5 μm, deparaffinized, stained with H&E and observed under an Aperio ScanScope (Leica ICC50 W, Wetzlar, Germany). For preparing frozen sections of mandibles, Kawamoto’s film method was performed [36]. Briefly, samples were embedded with Super Cryoembedding Medium (SECTION-LAB Co. Ltd., Hiroshima, Japan) and cut to a thickness of 3 μm with a tungsten carbide blade after mounting the adhesive film onto the cut surface. For immunohistochemical staining, the specimens were fixed with 4% PFA for 10 min, followed by incubation with primary antibodies at 4 °C overnight after blocking with 10% normal donkey serum (Jackson Immunoresearch, West Grove, PA, USA) for 60 min at room temperature. Primary antibody specific to Type VI collagen (Fitzgerald, North Acton, MA, USA) was applied to samples at a 1:50 dilution. After washing, the specimens were incubated with secondary antibody Alexa Fluor 647 anti-rabbit at a 1:500 dilution (Thermo Fisher Scientific, Waltham, MA, USA) for 60 min at room temperature. As a negative control PBS was applied instead of primary antibodies. All images were taken by fluorescence microscope.

### 4.4. Protein Extraction

Alveolar bone was dissected from 6-week-old WT and *Col6a2*-KO mice with soft tissues attached to the bone removed and immediately snap-frozen in liquid nitrogen. Alveolar bone was subsequently stored at −80 °C until protein extraction. The samples were then put into the center of a tissue tube (Covaris, Woburn, MA, USA) frozen in liquid nitrogen, and pulverized on the CP02 cryoPREP Automated Dry Pulverizer (Covaris, Woburn, MA, USA). Pulverized samples were processed using a buffer (100 µL of 4 M Guanidine-HCL and 100 µm of 0.25 M EDTA) containing protease and phosphatase inhibitors for 48 h and then the buffer was exchanged after centrifugation to collect the supernatant and incubated with the same buffer for another 48 h. Protein extracts were precipitated with 4 volumes of acetone, and protein pellets were resuspended with 5 mM triethylammonium bicarbonate buffer (TEAB). Protein concentrations were determined using the Bradford assay (Thermo Fisher Scientific, Waltham, MA, USA). Protein extracts were then reduced with 50 mM tris 2-carboxyethyl phosphine (TCEP) and alkylated with 10 mM iodoacetamide (IAA). Trypsin digestion was performed with 1 µg of MS grade trypsin (Promega, Madison, WI, USA) to 25 µg of sample proteins at 37 °C over night. The resulting peptide was desalted using C18 Toptip columns (Glygen Scientific Corp, Columbia, MD, USA) following manufactural protocol.

### 4.5. Proteomics

NanoLC-MS/MS analysis of tryptic peptides was carried out with a Thermo Scientific Fusion Lumos tribrid mass spectrometer interfaced to a UltiMate3000 RSLCnano HPLC system. For each analysis, 1 µg of the tryptic digest was loaded and desalted in an Acclaim PepMap 100 trapping column (75 µm × 2 cm) at 4 µL/min for 5 min. Peptides were then eluted into Thermo Scientific Accalaim PepMap™ 100 column, (3 µm, 100 Å, 75 µm × 250 mm) and chromato-graphically separated using a binary solvent system consisting of A: 0.1% formic acid and B: 0.1% formic acid and 80% acetonitrile at a flow rate of 300 nL/min. A gradient was run from 1% B to 37.5%B over 120 min, followed by a 5 min wash step with 80% B and 10 min equilibration at 1% B before the next sample was injected. Precursor masses were detected in the Orbitrap at R = 120,000 (*m*/*z* 200). Fragment masses were detected in linear ion trap at unit mass resolution. Data dependent MSMS was carried with the top of speed setting, cycle time was 2 s with dynamic exclusion of 20 s. Protein identification was carried out using Proteome Discoverer software package (v 2.5 Thermo Scientific). Raw data was searched against a mouse protein database from Uniprot along with a contaminant protein database with Sequest HT search engine. C carbamidomethylation was set as fixed modification, M oxidation, and protein N-terminal acetylation were set as variable modifications. Peptide precursor intensity-based label free quantification was carried out in Proteome Discoverer using unique and razor peptides. Abundances from different samples were normalized by total peptide amount from each sample, Protein abundances were calculated by summed intensity of top 3 peptides.

### 4.6. Ligature-Induced Periodontitis Model

Periodontal inflammation was induced by ligation of the maxillary second molar in mice, as previously described [37]. A 5-0 silk ligature (Corza medical, Westwood, MA, USA) was tied around the maxillary right second molar in 11-week-old mice under anesthesia. The contralateral second molar in each mouse was un-ligated to serve as baseline control for bone height and volume measurements. The mice were euthanized 3 days after surgery and subjected to imaging with µCT using the same setting as was done with analysis of the mandibles. The ligatures remained in place in all mice throughout the experimental period.

### 4.7. Statistics

Statistical analyses were performed with unpaired Student’s *t*-test. A statistically significant difference was considered as *p*-value < 0.05.

## Figures and Tables

**Figure 1 ijms-23-14347-f001:**
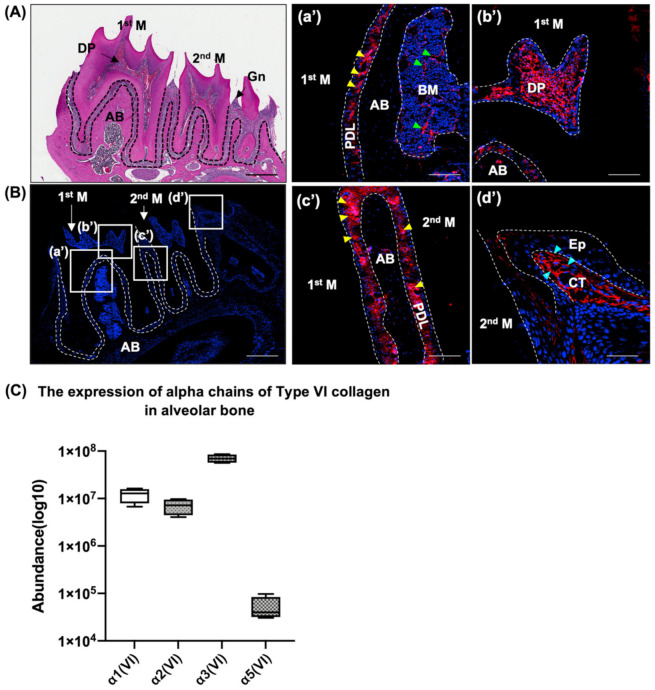
The expression of Type VI collagen in oral tissues. (**A**) H&E staining of mandibles from 3-month-old mice. Dashed line shows periodontal ligament. (**B**) Representative immunofluorescent staining image of Type VI collagen in mouse mandibles of 3-month-old mice (*n* = 3). Magnified images of boxed areas in panel B are shown in (**a’**,**b’**,**c’**,**d’**). Nuclei were stained with DAPI (blue). Type VI collagen expression (red) in the periodontal ligament (PDL) (yellow arrowheads), bone marrow (BM) (green arrowheads), gingiva (Gn (blue arrowheads). (**C**) The abundance of alpha chains of Type VI collagen in murine alveolar bone from 6-week-old mice. M: molar, AB: alveolar bone, DP: dental pulp, Gn: gingiva, BM: bone marrow, PDL: periodontal ligament, Ep: epithelium, CT: connective tissue. Scale bars: (**A**,**B**): 500 µm, (**a’**–**d’**): 50 µm.

**Figure 2 ijms-23-14347-f002:**
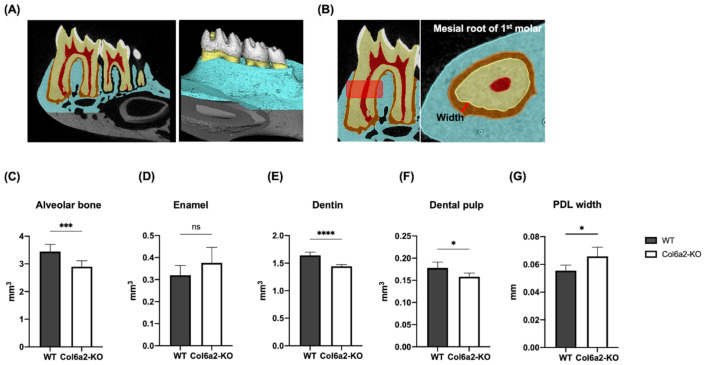
µCT analysis of the volume and width of the PDL. (**A**) An illustration showing separate analysis of each structure in the oral tissues. Each color represents a different tissue: enamel (white), dentin (yellow), dental pulp (red), PDL (orange) and alveolar bone (blue). (**B**) shows the width of the PDL measured by µCT. Quantification of the different tissues is shown for alveolar bone (**C**), enamel (**D**), dentin (**E**), dental pulp (**F**). (**G**) The PDL width was measured in the mesial root of the 1st molar. Bars represent standard deviation (volume: *n* = 7/genotype, PDL width: *n* = 5/genotype) *p*-value: * *p* < 0.05, *** *p* < 0.005 **** *p* < 0.001 ns: not significant.

**Figure 3 ijms-23-14347-f003:**
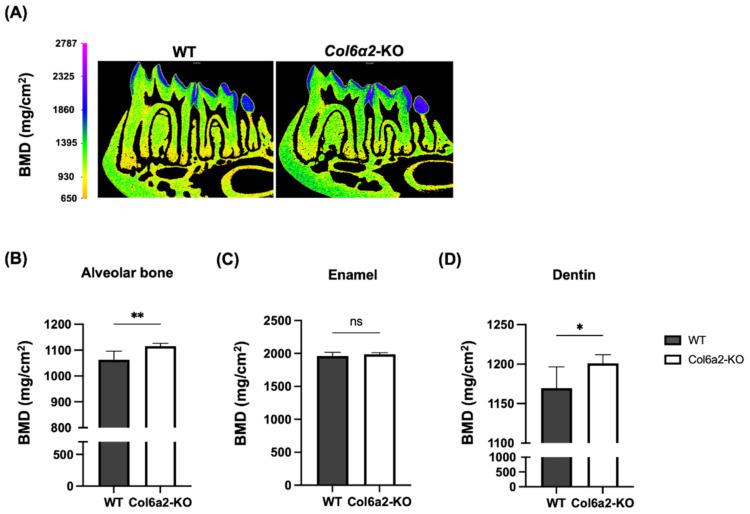
Mineral density with µCT analysis. (**A**) Heatmap of mineral density from WT and Col6α2-KO mice mandibles. Quantification of the data is shown in (**B**): Alveolar bone, (**C**): Enamel and (**D**): Dentin. Bars represent standard deviation (*n* = 7/genotype) *p*-value: * *p* < 0.05, ** *p* < 0.01 ns: not significant.

**Figure 4 ijms-23-14347-f004:**
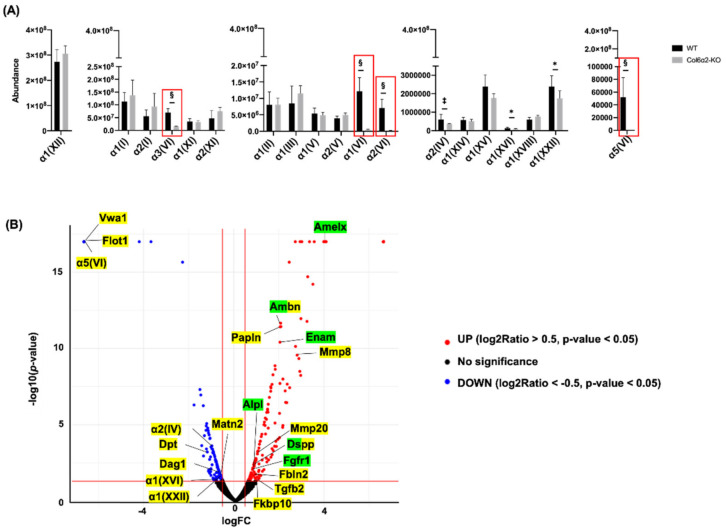
Proteome analysis with protein extracted from alveolar bone at 6-week of age. (**A**) The abundance of collagen detected in alveolar bone. (**B**) The volcano plot showing deferentially expressed proteins in Col6α2-KO mice alveolar bone. Green marked proteins represent molecules related to Biomineralization and yellow marked proteins represent molecules related to Extracellular matrix organization. Bars represent standard deviation (*n* = 4/genotype) *p*-value: * *p* < 0.05, ‡ *p* < 0.005, § *p* < 0.001.

**Figure 5 ijms-23-14347-f005:**
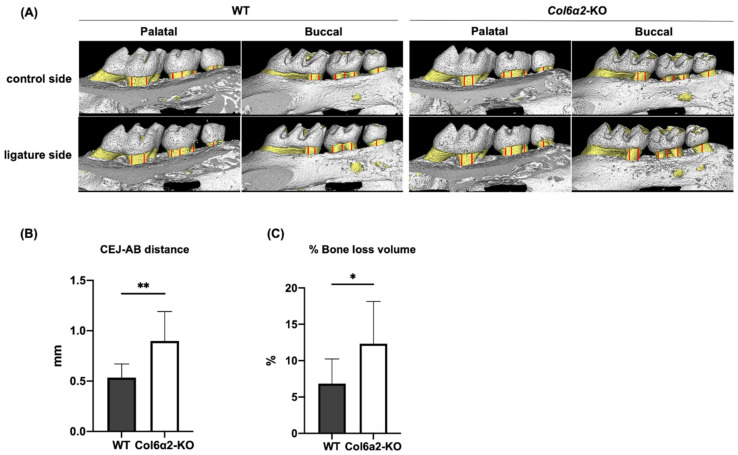
Alveolar bone loss with periodontitis. (**A**) Representative 3D reconstructed image of left molars in maxilla (control side) and right molars in maxilla (ligature side). To evaluate bone loss, the distance between the cemento-enamel junction (CEJ) to the alveolar bone crest (**A**,**B**) was measured at 12 separate points in both the buccal and palatal sides. The red line in (**A**) shows the measurement points. (**B**) shows the difference of the CEJ-AB distance in the control side and the ligature side from WT and *Col6α2*-KO mice. (**C**) shows the % volume of bone loss. Bars represent standard deviation (*n* = 9/genotype) *p*-value: * *p* < 0.05, ** *p* < 0.01.

## Data Availability

Data available in a publicly accessible repository. The data presented in this study are openly available in [PRIDE (https://www.ebi.ac.uk/pride/, accessed on 14 October 2022)] with accession PXD037422.

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
