# Peer review of "The Role of Type VI Collagen in Alveolar Bone"

_ijms, 2022, doi:10.3390/ijms232214347_

Round 1

Reviewer 1 Report

1.     In Figure 2, the authors compared the volume difference for each tooth component. Has there been any consideration of the volume difference between the overall tooth and the alveolar bone?

2.     Clo6a2-KO mice are deficient in type IV collagen. Bone growth and metabolism are thought to be low compared to normal (WT), but what do you think is the reason for the higher BMD?

3.     The Micro CT ROI setting part is not well understood. Does this mean that the ROI is set between 25 slices below the CEJ and 25 slices above the root apex as a boundary?

4.     In experiments for protein extraction, the authors used 6-week-old mice. In discussion section, the authors explained that the reason why mice aged 3 months were selected was oral tissues were completed at that time. These two parts seem to contradict each other. Why did the authors select 6-week-old mice for protein extraction?

5.     Why was immunohistochemistry performed only on WT? Analysis using Micro-CT is also not meaningless, but if there is a difference in type IV collagen, it is also meaningful to perform IHC in both groups.

6.     In the #37 reference cited by the authors, the palatal side shows a difference from the control after 3 days of ligature, but the buccal side shows a difference after 5 days. Is there any special reason to sacrifice after 3 days of ligature?

7.     It is not clearly understood the part below that the authors describe in the discussion.

1)     When we analyzed the publicly available data bases from this study, we found 265 that Type VI collagen expression was upregulated in gingiva from periodontitis patients 

 According to the authors' study, it is understood that knock-out mice are susceptible to periodontitis due to weakened type IV collagen expression. The two seem to conflict, but I'd like an explanation.

2)     Type VI collagen could be a factor 271 to consider in developing a new therapy for treating or preventing periodontitis.

 In addition, the current study is thought to be that if there is a problem with type IV collagen, it becomes vulnerable to periodontitis, but whether the opposite case can be established is another question. It seems that there is a logical leap forward to say that type IV collagen is a new treatment for preventing and treating periodontitis. Please consider this point.

Author Response

Reviewer’s comment:

  1. In Figure 2, the authors compared the volume difference for each tooth component. Has there been any consideration of the volume difference between the overall tooth and the alveolar bone?

Authors response:

We also wondered about the possible effect of Type VI collagen depletion on the relative difference in volume between the overall tooth and the alveolar bone. When we measured these parameters, we found there was no significant difference in this ratio. To clarify this point, we added the statement “There were no significant differences in the ratio in volume of the overall tooth to alveolar bone volume in Col6a2-KO compared to WT mice.”

  1. Clo6a2-KO mice are deficient in type IV collagen. Bone growth and metabolism are thought to be low compared to normal (WT), but what do you think is the reason for the higher BMD?

Authors response:

The reviewer raises and interesting point regarding the possible relationship between low bone growth and metabolism and, further, what this might mean in the context of higher BMD in the alveolar bone. Based on our analysis of body size we found Col6α2-KO mice are statistically smaller than age matched WT mice and that bone resporption and not bone growth was higher in the long bones of the Col6α2-KO compared to WT mice (Pham, H. T et al., Scientific Reports 2020). Regarding the mechanisms causing higher BMD in the alveolar bone, we will need further studies, however, it is suggested that type IV collagen is important for the regulation of ECM organization, and that might affect biomineralization. From our current data, we speculate that upregulated proteins related to biomineralization including Enam, Ambn, Amelx, Dspp, Alpl and Fgfr1 is the mechanistic basis for the changes we see bone mineralization in the alveolar bone. We describe our ideas regarding high BMD in alveolar bone in the Discussion. (230-249).

  1. The Micro CT ROI setting part is not well understood. Does this mean that the ROI is set between 25 slices below the CEJ and 25 slices above the root apex as a boundary?

Authors response:

We agree our description of this analytical method was confusing. We revised the paper to clarify how we made our measurements as follows: “To measure the PDL width, a region of interest (ROI) was determined using the cemento-enamel junction (CEJ) and the apex of the root. The middle position of root was calculated from the CEJ and the apex of the root, and 25 slices from middle position to CEJ side and 25 slices from middle position to the apex side was used for the ROI.”

  1. In experiments for protein extraction, the authors used 6-week-old mice. In discussion section, the authors explained that the reason why mice aged 3 months were selected was oral tissues were completed at that time. These two parts seem to contradict each other. Why did the authors select 6-week-old mice for protein extraction?

Authors response:

We regret our failure to explain why we used 6-week-old mice for protein extraction. To address this point, we added the statement “In mice teeth, cementum tissue covers root deposits with aging, making it difficult to pull out the tooth intact. To avoid contamination from other tooth tissues, we used this age of mice.” (in the result 2.1 section). Further, the use of young mice was reasonable because it allowed us to examine a developmental stage active in mineral apposition. We revised the result 2.3 section to clarify this point.

  1. Why was immunohistochemistry performed only on WT? Analysis using Micro-CT is also not meaningless, but if there is a difference in type IV collagen, it is also meaningful to perform IHC in both groups.

Authors response:

We believe the reviewer wants us to emphasize the important networking role Type VI collagen (COL6). In this regard the reviewer notes, we found less expression of Type IV collagen (COL4) in Col6α2-KO mice in our proteomics analysis and the reviewer would like us to follow up with IHC. It is known that COL6 binds to COL4, and that it could be a factor affecting ECM organization. We also think it is interesting to examine the way COL6 works as a networking factor with other ECM molecules like COL4 and, further, how this could affect mineral density, however, we believe it is out of scope for this first report on the role of COL6 on oral tissue structure.

  1. In the #37 reference cited by the authors, the palatal side shows a difference from the control after 3 days of ligature, but the buccal side shows a difference after 5 days. Is there any special reason to sacrifice after 3 days of ligature?

Authors response:

The method to induce periodontitis is widely used, however, the amount of bone resorption can vary widely depending on the individuals who perform the surgery, so we needed to optimize the time point for ultimate evaluation in our own hands. To determine the optimal time for this, we measured bone resorption multiple days after surgery and found there was significant difference of bone resorption between WT and Col6α2-KO mice in both the buccal and on the palatal side at 3 days, so we chose that time point to collect data that is presented in our paper.

  1. It is not clearly understood the part below that the authors describe in the discussion.

 1) When we analyzed the publicly available data bases from this study, we found that Type VI collagen expression was upregulated in gingiva from periodontitis patients. According to the authors' study, it is understood that knock-out mice are susceptible to periodontitis due to weakened type IV collagen expression. The two seem to conflict, but I'd like an explanation.

Authors response:

We acknowledge we did not provide enough explanation about this. To clarify our interpretation of the data, we added the statement “When we analyzed the publicly available data bases from this study, we found that Type VI collagen expression was upregulated in the gingiva from periodontitis patients, suggesting Type VI collagen is secreted to protect tissues from inflammation or to induce regeneration of tissues subject to destruction. (Data not shown).”

 2) Type VI collagen could be a factor to consider in developing a new therapy for treating or preventing periodontitis. In addition, the current study is thought to be that if there is a problem with type IV collagen, it becomes vulnerable to periodontitis, but whether the opposite case can be established is another question. It seems that there is a logical leap forward to say that type IV collagen is a new treatment for preventing and treating periodontitis. Please consider this point.

Authors response:

We agree with reviewer’s point that our proposal about the potential therapeutic value of Type VI collagen in periodontitis is overreaching. For this reason, we deleted the statement regarding Type VI collagen as a new therapy target in the discussion.

Reviewer 2 Report

The authors used an α2(VI) deficient mouse (Col6α2-KO) model to examine the role of Type VI collagen in oral tissues and found the cues between type VI collagen deficiency and periodontitis with the Ligature-induced periodontitis model. The manuscript needs  minor revision.

1.Please specify the number of knockout/WT mice used in this study and how many mice used for imaging analysis or proteome analysis.

2.It was shown in your article that both Col6α2-KO mice and fibromodulin knock-out mice had greater mineral density in alveolar bone(lines 202-209).  The authors speculated that it may be related to phosphate regulation.  So did the rest of the skeletal bone mineral density profile of these knockout mice also go up?

3.Please specify the age of the mice in the legend of Figure 1 C and Figure 4

4.Why use 6-week-old mice in proteomics analysis instead of 3-month-old mice?

5.As the results in Figure 4, the important molecules of “Biomineralization”  were upregulated in α2(VI) deficient alveolar bone.  Can the authors perform western blot for the marker molecules upregulated in the alveolar bone? This would support the data of proteome results.

6.As described in the article, type VI collagen expression was upregulated in gingiva from periodontitis patients(Line 263-270). How about type VI collagen expression in the periodontal ligaments of patients with periodontitis?

Author Response

Reviewer 2

Reviewer’s comment:

  1. Please specify the number of knockout/WT mice used in this study and how many mice used for imaging analysis or proteome analysis.

Authors response:

To address this point, we added the statement “40 WT and Col6a2-KO were used in this study.” in section 4.1 and also added the exact number of mice used for imaging and proteomics.

  1. It was shown in your article that both Col6α2-KO mice and fibromodulin knock-out mice had greater mineral density in alveolar bone (lines 202-209). The authors speculated that it may be related to phosphate regulation. So, did the rest of the skeletal bone mineral density profile of these knockout mice also go up?

Authors response:

Unfortunately, we didn’t analyze other bones in the fibromodulin knock-out mice, and we don’t have this mouse model as a living colony because we had to freeze them down during the COVID pandemic. As for Col6α2-KO mice, we observed decreased BMD in the vertebrae and femur, which interestingly is the opposite of what we found in the alveolar bone. Our interpretation regarding these tissue specific differential outcomes is described in 250 – 260.

  1. Please specify the age of the mice in the legend of Figure 1 C and Figure 4

Authors response:

We thank the reviewer for catching this mistake. We added the age of mice in both.

  1. Why use 6-week-old mice in proteomics analysis instead of 3-month-old mice?

Authors response:

We acknowledge there was not enough explanation regarding the different ages used in the study. We added statement to clarify our rationale that is described for reviewer 1 above and in the revised paper.

  1. As the results in Figure 4, the important molecules of “Biomineralization” were upregulated in α2(VI) deficient alveolar bone. Can the authors perform western blot for the marker molecules upregulated in the alveolar bone? This would support the data of proteome results.

Authors response:

We agree that western blotting would be useful for future studies aimed to confirm the data we found from the proteomics analysis. We added the statement “An important next step will be to carry out western blotting to confirm there is upregulation of specific proteins related to biomineralization including SIBLINGs and EMPs.” in the discussion (248 - 249).

6.As described in the article, type VI collagen expression was upregulated in gingiva from periodontitis patients (Line 263-270). How about type VI collagen expression in the periodontal ligaments of patients with periodontitis?

Authors response:

We thank the reviewer for this insightful comment. Type VI collagen is an understudied molecule in inflammation research, and there are few reports showing its role in the PDL. To understand the role of Type VI collagen in the PDL in periodontitis, we are currently establishing methods to collect PDL cells from mice teeth. Our next plan is to analyze the population of cells in the PDL subjected to induced periodontitis with focus on immune cells and fibroblasts and well as the expression of Type VI collagen in this process. In summary, at this time, there is no data showing the expression of Type VI collagen in PDL in human periodontitis patients, but we acknowledge this will be important to determine in future investigations.